# COVID-19 mortality rate and its associated factors during the first and second waves in Nigeria

Kelly Elimian[1,2,3,4]*, Anwar Musah[4,5], Carina King[2], Ehimario Igumbor[1,4,5], Puja Myles[6], Olaolu Aderinola[1], Cyril Erameh[7], William Nwanchukwu[1], Oluwatosin Akande[1], Ndembi Nicaise[8], Oladipo Ogunbode[1], Abiodun Egwuenu[1], Emily Crawford[1], Giulia Gaudenzi[2], Ismail Abdus-Salam[9], Olubunmi Olopha[1], Yahya Disu[1], Abimbola Bowale[10], Cyprian Oshoma[3], Cornelius Ohonsi[1], Chinedu Arinze[1], Sikiru Badaru[1], Blessing Ebhodaghe[1], Zaiyad Habib[11], Michael Olugbile[12], Chioma Dan-Nwafor[1], Jafiya Abubakar[1], Emmanuel Pembi[13], Lauryn Dunkwu[14], Ifeanyi Ike[1,15], Ekaete Tobin[7], Bamidele Mutiu[16], Rejoice Luka-Lawal[1], Obinna Nwafor[1], Mildred Okowa[17], Chidiebere Ezeokafor[4,18], Emem Iwara[19], Sebastian Yennan[1], Sunday Eziechina[1], David Olatunji[1], Lanre Falodun[20], Emmanuel Joseph[21], Ifeanyi Abali[1], Tarik Mohammed[1], Benjamin Yiga[22], Khadeejah Kamaldeen[23], Emmanuel Agogo[24], Nwando Mba[1], John Oladejo[1], Elsie Ilori[1], Olusola Aruna[25], Geoffrey Namara[26], Stephen Obaro[27], Khadeejah Hamza[4,28], Michael Asuzu[4,29], Shaibu Bello[4,30], Friday Okonofua[31], Yusuf Deeni[4,32], Ibrahim Abubakar[33], Tobias Alfven[2], Chinwe Ochu[1], Chikwe Ihekweazu[1]

1 Nigeria Centre for Disease Control, Abuja, Nigeria, 2 Department of Global Public Health, Karolinska Institutet, Stockholm, Sweden, 3 Department of Microbiology, Faculty of Life Sciences, University of Benin, Benin City, Edo State, Nigeria, 4 Nigeria COVID-19 Research Coalition, Abuja, Nigeria, 5 Department of Geography, University College London, London, United Kingdom, 6 Clinical Practice Research Datalink, Medicines and Healthcare Products Regulatory Agency, London, United Kingdom, 7 Institute of Lassa Fever Research and Control, Irrua Specialist Teaching Hospital, Irrua, Edo State, Nigeria, 8 Africa Centres for Disease Control and Prevention, Addis-Ababa, Ethiopia, 9 Lagos State Ministry of Health, Lagos, Lagos State, Nigeria, 10 Infectious Disease Unit, Mainland Hospital, Lagos, Lagos State, Nigeria, 11 University of Abuja Teaching Hospital, Abuja, Nigeria, 12 The World Bank, Abuja, Nigeria, 13 Adamawa State Ministry of Health and Human Services, Yola, Adamawa State, Nigeria, 14 Tony Blair Institute for Global Change, Abuja, Nigeria, 15 eHealth Africa, Abuja, Nigeria, 16 Lagos State Biobank Mainland Hospital Yaba, Lagos, Lagos State, Nigeria, 17 Ministry of Health, Asaba, Delta State, Nigeria, 18 National Agency for the Control of AIDS, Abuja, Nigeria, 19 Maryland Global Initiatives Corporation, Abuja, Nigeria, 20 Department of Internal Medicine, National Hospital, Abuja, Nigeria, 21 Kaduna State Infectious Disease Control Centre, Kaduna, Kaduna State, Nigeria, 22 Bauchi State Ministry of Health, Bauchi, Bauchi State, Nigeria, 23 Kwara State Ministry of Health, Ilorin, Kwara State, Nigeria, 24 Resolve to Save Lives, Abuja, Nigeria, 25 International Health Strengthening Project, Global Public Health, Public Health England, Abuja, Nigeria, 26 World Health Organization, Abuja, Nigeria, 27 Department of Paediatrics, University of Nebraska, Lincoln, Nebraska, United States of America, 28 Department of Community Medicine, Ahmadu Bello University, Zaria, Nigeria, 29 University College Hospital, Ibadan, Oyo State, Nigeria, 30 College of Health Sciences, Usmanu Danfodiyo University, Sokoto, Sokoto State, Nigeria, 31 Centre of Excellence in Reproductive Health Innovation, University of Benin, Benin City, Edo State, Nigeria, 32 Department of Microbiology and Biotechnology, Faculty of Science, Federal University Dutse, Dutse, Jigawa State, Nigeria, 33 Institute for Global Health, Faculty of Pop Health Sciences, University College London, London, United Kingdom

* Kelly.elimian@ki.se

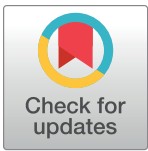

**Data Availability Statement:** The dataset utilised for this study can be made available upon reasonable request to the Head of NCDC Research at chinwe.ochu@ncdc.gov.ng.

## Abstract

COVID-19 mortality rate has not been formally assessed in Nigeria. Thus, we aimed to address this gap and identify associated mortality risk factors during the first and second

**Funding:** The authors received no specific funding for this work.

**Competing interests:** The authors have declared that no competing interests exist.

waves in Nigeria. This was a retrospective analysis of national surveillance data from all 37 States in Nigeria between February 27, 2020, and April 3, 2021. The outcome variable was mortality amongst persons who tested positive for SARS-CoV-2 by Reverse-Transcriptase Polymerase Chain Reaction. Incidence rates of COVID-19 mortality was calculated by dividing the number of deaths by total person-time (in days) contributed by the entire study population and presented per 100,000 person-days with 95% Confidence Intervals (95% CI). Adjusted negative binomial regression was used to identify factors associated with COVID-19 mortality. Findings are presented as adjusted Incidence Rate Ratios (aIRR) with 95% CI. The first wave included 65,790 COVID-19 patients, of whom 994 (1·51%) died; the second wave included 91,089 patients, of whom 513 (0·56%) died. The incidence rate of COVID-19 mortality was higher in the first wave [54·25 (95% CI: 50·98–57·73)] than in the second wave [19·19 (17·60–20·93)]. Factors independently associated with increased risk of COVID-19 mortality in both waves were: age ≥45 years, male gender [first wave aIRR 1·65 (1·35–2·02) and second wave 1·52 (1·11–2·06)], being symptomatic [aIRR 3·17 (2·59–3·89) and 3·04 (2·20–4·21)], and being hospitalised [aIRR 4·19 (3·26–5·39) and 7·84 (4·90–12·54)]. Relative to South-West, residency in the South-South and North-West was associated with an increased risk of COVID-19 mortality in both waves. In conclusion, the rate of COVID-19 mortality in Nigeria was higher in the first wave than in the second wave, suggesting an improvement in public health response and clinical care in the second wave. However, this needs to be interpreted with caution given the inherent limitations of the country's surveillance system during the study.

## Introduction

Nigeria implemented a series of public health interventions following the declaration of the COVID-19 pandemic on March 11, 2020, amidst a fragile and under-resourced healthcare system complicated by economic, political, social, and security challenges [1]. For example, before and during the detection of the first few cases of COVID-19 in Nigeria, several public health measures were instituted by the government and its partners, including strengthening in-country molecular diagnostic capacity, equipping infectious disease treatment centres for case management, and training of healthcare and allied-healthcare workers [1]. In addition to adopting a co-production approach—which allowed a multidisciplinary stakeholder—to collaborate in addressing issues arising in real-time [2], the country invested in developing and revising guidelines for surveillance, case management and infection prevention and control [1].

As of October 16th 2021, Nigeria, with a population of over 200 million people, had tested total samples of 3,142,971 for COVID-19, recording a cumulative number of 208,797 confirmed cases and 2,769 deaths, with a case fatality ratio (CFR, %) of 1·3% [3]. During the same period, the cumulative number of confirmed COVID-19 cases in the World Health Organization (WHO) African region was 6,009,444, with 147,749 deaths and a CFR of 2·5% [4]. Whilst a robust surveillance of COVID-19 transmission requires that countries conduct widespread testing, evidence from the inception of the pandemic to date indicates that Africa has the least COVID-19 cases and testing numbers compared to other regions [5]. Regionally, as of June 11, 2020, Nigeria had the highest absolute numbers of COVID-19 and associated mortality of any country in West Africa [6]. This scenario was attributed mainly to the country's large

population density and international air traffic [6]. In April 2021, the second wave of the pandemic in Nigeria was declining, with COVID-19 cases and related mortalities reported through the Nigeria Centre for Disease Control (NCDC) surveillance consistently at less than 30 and 5 per day, respectively.

Findings from independent systematic reviews and meta-analyses [7, 8] indicate that older age, male gender, current smoker, pre-existing comorbidities, dyspnoea, complications during hospitalisation, and corticosteroid therapy significantly increase the risk of COVID-19 mortality. Similar to global evidence, we found older age (≥51 years) and presentation with cough, breathing difficulties and vomiting to significantly increase the risk of COVID-19 mortality in Nigeria [9]. Notably, CFR is commonly used to describe mortalities associated with COVID-19; however, the estimates may vary depending on the case definition–total cases (all COVID-19 cases) or only confirmed cases [10]. Classifying cases as 'recovered' is particularly challenging considering long-COVID, where a range of symptoms can persist for months. Thus, the estimated CFR in Nigeria and countries with similar surveillance capacity and sociodemographic profiles could be overestimated as the denominator includes a subset of all COVID cases [11]. Moreover, determining the denominator for CFR estimation can be difficult as asymptomatic patients at testing or those with very mild symptoms might not be tested and missed by the surveillance system [12]. For example, we found that up to 66% of 12,289 confirmed COVID-19 cases in Nigeria between February and June 2020 were asymptomatic at testing [13]. This is despite the fact that asymptomatic contacts of confirmed COVID-19 cases meeting the NCDC testing criteria [14] are eligible for testing in line with the country's testing strategy. Furthermore, evidence from post-mortem surveillance of deceased persons within 48 hours of death in Zambia suggests that a substantial number of COVID-19 deaths, especially those at the community level, could be missed due to low testing [15]. Thus, CFR estimates in settings such as Nigeria could be biased.

A comparison of mortality incidence rates in the two waves of the pandemic will improve understanding of the context-specific dynamics of the disease and its outcomes in Nigeria, and possibly shed more light on the severity of the pandemic. Furthermore, identifying the factors associated with COVID-19 mortalities will facilitate the identification of vulnerable population groups to inform public health and clinical management strategies. Therefore, this study estimated the incidence rate of COVID-19 mortalities and investigated the socio-demographic and clinical characteristics associated with COVID-19 mortality during the first and second waves in Nigeria.

## Methods

### Ethics statement

The protocol for this study was reviewed and approved by the Nigeria National Health Research Ethics Committee (NHREC/01/01/2007-22/06/2020). The analysed data were fully anonymised before being accessed from the NCDC Surveillance and Epidemiology Department. Being an analysis of secondary data, informed consent was not sought but all research activities were conducted in line with the ethical approval.

### Study design and settings

This was a retrospective analysis of national surveillance data collected from all 36 States and the Federal Capital Territory (FCT) in Nigeria. The three tiers of healthcare systems in Nigeria (primary, secondary, and tertiary) and designated areas (e.g., State Ministry of Health and *ad hoc* testing centres) served as COVID-19 testing centres. Although secondary and tertiary hospitals formally serve as treatment centres for COVID-19 patients in Nigeria, NCDC guidelines

[16] allow health workers to manage low-risk patients (e.g., <60 years with no history of non-communicable disease, asymptomatic or with mild symptoms etc.) at home.

## Study population

The study population included individuals (symptomatic and asymptomatic at testing) who tested positive for SARS-CoV-2 by Reverse-Transcriptase Polymerase Chain Reaction (RT-PCR) [17]. Eligibility for an RT-PCR test was based on an individual meeting the NCDC COVID-19 suspect case definition [14] used during the first and second waves. However, regardless of symptomatic status, persons who had close contact with a confirmed or probable COVID-19 case were tested. The identification of such persons was made during contact tracing or, based on personal concerns over one's health, presentation to a testing centre.

## Data collection and management

All Nigerian States and their respective local government areas are required to actively monitor and report infectious diseases of public health importance to NCDC via the Integrated Disease Surveillance and Response (IDSR) system [18]. All surveillance data within NCDC are collected using the Surveillance Outbreak Response Management and Analysis System (SORMAS) database. SORMAS is a module-based open-source real-time electronic surveillance database with mobile and web application packages. The data used in this study are all COVID-19 records (positive and negative SARS-CoV-2 tests) within SORMAS from February 27, 2020, to April 3, 2021.

A detailed description of data collection is available in a previous paper [13]. Briefly, trained healthcare personnel assessed suspected COVID-19 cases who met the NCDC case definitions [14] during a patient's visit to a testing centre or via contact tracing. Healthcare personnel completed electronic case investigation forms s containing demographic and clinical information and collected a minimum of one nasopharyngeal or nasal swab and one oropharyngeal swab for laboratory diagnosis. Clinical outcomes for patients with COVID-19 are updated on SORMAS regularly as "recovered," "dead," or "currently ill" by designated persons (e.g., clinicians, State Epidemiologists, Disease Surveillance and Notification Officer (DSNO) or SORMAS Implementation Officers). The continuous update of SORMAS is facilitated by reviewing patient records (treatment centres and homes) and, where possible, contact tracing in the community. This process is facilitated by the feedback mechanism within the IDSR system, such that death at the local government level can be identified and reported on SORMAS by a DSNO. However, deaths in the community could be missed by the surveillance system.

The outcome variable was mortalities among persons who tested positive for SARS-CoV-2, irrespective of a cause of death being assigned or not, within 30 days of sample collection for laboratory diagnosis. We defined survivors as COVID-19 patients who were discharged from a health facility as per the NCDC discharge criteria in use during the study period (or cleared by a healthcare worker for those who received home care), or patients lost to follow-up up within 30 days of sample collection for COVID-19 diagnosis. For example, asymptomatic patients at testing in June 2020 onwards were discharged 14 days after the initial positive result (e sample collection date) [19].

The start date for the first wave was February 27, 2020, the date the COVID-19 index case was confirmed in Nigeria (corresponding to the epidemiologic week 9) and ended on October 24, 2021. The second wave started on October 25, 2020 (epidemiologic week 44), but the study ended on April 3, 2021 (epidemiological week 13)—the date when the analysed dataset was extracted from SORMAS. Each person entered the study ("entry date") and thus contributed person-time on the date that their initial sample for COVID-19 diagnosis was taken as

registered on SORMAS. Individuals exited this study: if they died within 30 days of the sample being taken (death date); if a COVID-19 patient was discharged from the health facility after recovery (discharge date); the study end date or the last date of data contribution for non-hospitalised survivors (i.e., right censored). We handled missing data using the missing indicator approach. Table 1 below presents the definitions of the study covariates.

## Statistical analyses

We calculated frequencies and percentages of SARS-CoV-2 positive persons who had died or survived by patients' characteristics. Incidence rates of COVID-19 mortality was calculated by dividing the number of deaths by total person-time (in days) contributed by the entire study population and presented per 100,000 person-days with 95% Confidence Intervals (95% CI). We plotted Kaplan-Meier curves to examine the survival patterns of COVID-19 patients and compare differences between patients' characteristics using log-rank tests.

In addition, maps were developed to illustrate the geographical burden of COVID-19 in Nigeria, as well as to demonstrate which areas either increased (or decreased) by comparing the rates between Wave 1 and Wave 2. The incidence and mortality rates of COVID-19 were estimated at a Local Government Authority-level (LGA). Due to data sparsity of census data relating to the populations' counts and composition in Nigeria for 2020 and 2021, we used population density estimates produced by the Worldpop.org (https://www.worldpop.org/).

**Table 1. Definition and classification of study covariates.**

| Variable‡ | Definition |
|---|---|
| Epidemiologic week | It started with the week ending on the first Saturday of January; subsequent weeks began on Sunday and ended on Saturday. The current study covered weeks 9 to 53 of 2020 and weeks 1 to 13 of 2021. |
| Epidemiologic wave | Epidemiological wave (herein: wave) was defined as the time from the start of a peak (first week with increasing numbers of cases) to the end of a peak (week with a nadir of cases before the subsequent rise). The wave was classified as the first wave (week 9–43 of 2020) and the second wave (week 44 of 2020-week 13 of 2021). |
| Age (years) | Based on self-reports by an individual patient or a relative, age was treated both as continuous and categorical variables, depending on the study priority. As a categorical variable, age was classified based on clinical relevance in our context: 0–17; 18–25; 26–35; 36–45; 46–55; 56–64; ≥65. |
| Sex | It was classified as either male or female based on patient self-report. |
| Geopolitical zone | They are classified as a categorical variable, according to their State composition: South-west (Ekiti, Lagos, Ondo, Osun, Ogun, and Oyo States); South-south (Akwa-Ibom, Bayelsa, Cross-River, Delta, Edo, and Rivers State); South-east (Abia, Anambra, Ebonyi, Enugu, and Imo States); North-central (Benue, FCT, Kwara, Kogi, Nassarawa, Niger, and Plateau States); North-west (Jigawa, Kebbi, Kaduna, Kano, Katsina, Sokoto, and Zamfara States); and North-east (Adamawa, Bauchi, Borno, Gombe, Taraba, and Yobe States). |
| Symptomatic status | It was classified as a binary variable: Asymptomatic (no expression of any signs and symptoms) and symptomatic (expression of at least one sign or symptom) at the point of sample collection and completion of case investigation form. Signs and symptoms were defined relative to 14 days before sample collection. Examples of COVID-19 signs and symptoms included fever (axillary temperature of 37·5˚C or higher.), cough, difficulty breathing, diarrhoea, headache, etc. |
| Hospitalisation | It was defined as the admission of a COVID-19 patient, either for isolation or clinical need due to severity of illness, to a formal health facility for at least one night. It was classified as a binary variable (yes/no). |
| Education at diagnosis | It was classified as a categorical variable in line with the Nigerian educational system: No formal education; nursery/primary; secondary; and tertiary. |

‡Comorbidity was not included in the study due to the high proportion of missing data.

Detailed gridded population density data at a 100m-by-100m resolution were downloaded from this resource [20]; the grid-cells for the raster were merged with the shapefile for Nigeria's LGA boundary and through spatial overlays, we aggregated the grids to the LGA boundaries to estimate the total population for each LGA. The aggregated information was treated as denominators in the calculations for incidence and death rates for COVID-19 expressed per 100,000. It is worth noting that the raster for 2021 population counts are currently unavailable and therefore, we used those created for 2020 to calculate rates in Wave 1 (2020/2021) and Wave 2 (2021) with the assumption that population sizes in 2020 and 2021 do not differ substantially. Base layers implemented for this analysis include the three political borders (i.e., the shape file for the country, and the shape file for the states and LGA of Nigeria). They were originally sourced from GRID3 Nigeria (https://grid3.gov.ng). All geospatial analysis were carried in RStudio. We used a negative binomial multivariable Poisson regression model to identify risk factors associated with COVID-19 mortality. Our rationale for implementing a negative binomial regression model in this scenario was that it is a much better model for handling over-dispersed count data (as in the case of mortality related to COVID-19) than the traditional Poisson model. In addition, the model was informed by the frequency distribution of death counts due to COVID-19, after aggregating the events by descriptive characteristic groups, yielded large frequencies of groups with zero counts [21]. The selection of potential risk factors for COVID-19 mortality was based on previous research [9], biological plausibility and the availability of data routinely collected on the SORMAS platform. After the unadjusted regression analyses, we identified variables for inclusion in the final multivariate negative binomial regression analyses, using a stepwise (backward) elimination procedure. The level of statistical significance for a variable to remain in the model was $p<0\cdot05$, obtained from the likelihood ratio test for categorical variables and Wald's test for binary variables. Results are expressed as Incidence Rate Ratios (IRR) with corresponding 95% CIs. All statistical analyses were carried out in Stata version 16 (Stata Corp. LP, College Station, TX, United States of America). This report is structured in adherence to the relevant STROBE statement (see S1 Checklist).

## Results

The first wave included 65,790 patients diagnosed with COVID-19, with 1,832,290 person-days contributed to analyses. The second wave included 91,089 patients diagnosed with COVID-19, with 2,673,142 person-days contributed to analyses (Fig 1).

### Background characteristics of COVID-19 patients in relation to clinical outcome

There were 994 deaths (1·51%; 994/65,790) in the first wave and 513 deaths (0·56%; 513/91,089) in the second wave (Table 2). There were more deaths among older persons in both first and second waves, with the highest proportions recorded among patients aged 65 years or older (34·00% of deaths in the first wave and 43·27% in the second wave). A higher proportion of deaths was recorded in males in both waves. Over half of recorded deaths in each wave were among patients presenting with at least one symptom at diagnosis. There were fewer deaths among hospitalised patients than in non-hospitalised patients in both waves, though the first wave had more healthy patients.

### Survival pattern of COVID-19 patients in the first and second waves

Figs 2 and 3 depict the survival of COVID-19 patients by selected demographic and clinical characteristics in the first (left) and second (right) waves. Overall, the Kaplan-Meier survival

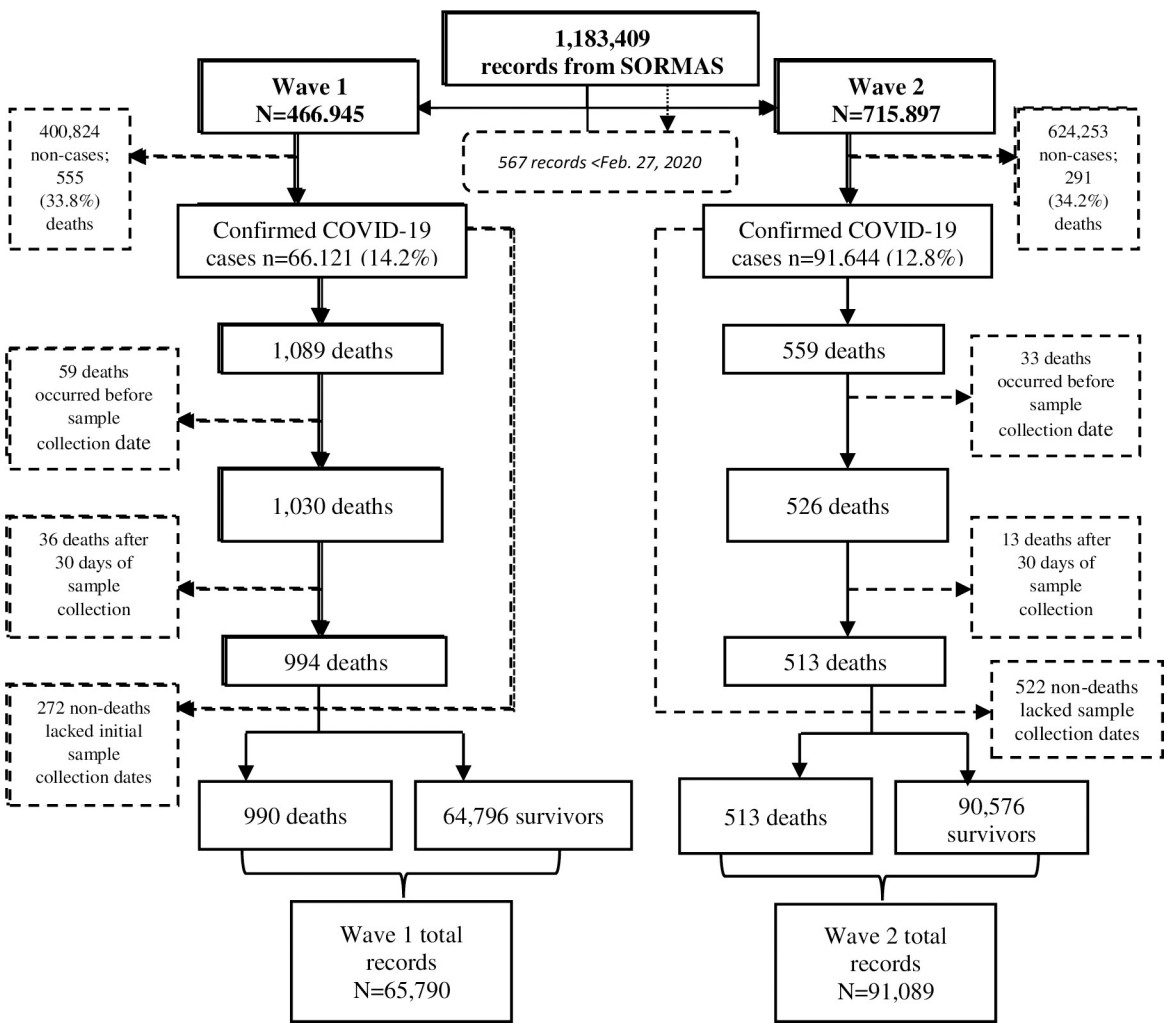

**Fig 1. A flow chart showing how the records, by wave, analysed for the study were selected.**

plots show that patients had worse survival in the first wave than in the second wave. This trend was reflected for all the sociodemographic and clinical characteristics explored in the present study.

## COVID-19 mortality rates in the first and second waves

The first wave had an overall mortality rate of 54·25 (50·98–57·73) per 100,000 person-days, while the second wave had a lower rate of 19·19 (17·60–20·93) per 100,000 person-days (Table 3). Mortality rates consistently increased with increasing age groups in both waves. The highest mortality rates were recorded in patients aged 65 years or older during the first wave [433·59; 389·75–482·37 per 100,000 person-days] and second wave [143·70; 125·98–163·90 per 100,000 person-days]. The mortality rate in male patients in the first wave was about twice as high as that of female patients [65·06 vs 37·16 per 100,000 person-days], and it remained higher in the second wave, albeit with a more marginal difference [22·18 vs 15·34 per 100,000 person-days]. Patients with secondary and tertiary education accounted for higher mortality rates in both waves. Regionally, the highest COVID-19 mortality rates in the first and second waves were recorded in the North-east [127·06; 102·98–156·77 per 100,000 person-days] and

**Table 2. Characteristics of COVID-19 patients, stratified by wave and clinical outcome.**

| Characteristic | Wave 1 | | | Wave 2 | | |
|---|---|---|---|---|---|---|
| | Survivor n = 64,796 (%) | Death n = 994 (%) | Total N = 65,790 | Survivor n = 90,576 (%) | Death n = 513 (%) | Total N = 91,089 |
| CFR* (%) | 1·51 | | | 0·56 | | |
| **Age group (year)** | | | | | | |
| 0–17 | 5,207 (8·04) | 15 (1·51) | 5,222 (7.94) | 8,037 (8·87) | 4 (0·78) | 8,041 (8.83) |
| 18–24 | 5,820 (11·44) | 21 (2·11) | 5,841 (8.88) | 9,291 (10·26) | 6 (1·17) | 9,297 (10.21) |
| 25–34 | 16,463 (25·41) | 51 (5·13) | 16,514 (25.10) | 22,731 (25·10) | 15 (2·92) | 22,746 (24.97) |
| 35–44 | 15,997 (24·69) | 103 (10·36) | 16,100 (24.47) | 20,014 (22·10) | 33 (6·43) | 20,047 (22.01) |
| 45–54 | 10,085 (15·56) | 174 (17·51) | 10,259 (15.59) | 14,030 (15·49) | 70 (13·65) | 14,100 (15.48) |
| 55–64 | 5,278 (8·15) | 257 (25·86) | 5,535 (8.41) | 8,095 (8·94) | 136 (26·51) | 8,231 (9.04) |
| ≥65 | 2,692 (4·15) | 338 (34·00) | 3,030 (4.61) | 5,213 (5·76) | 222 (43·27) | 5,435 (5.97) |
| Missing | 3,254 (5·02) | 35 (3·52) | 3,289 (5.00)‡ | 3,165 (3·49) | 27 (5·26) | 3,192 (3.50)‡ |
| **Sex** | | | | | | |
| Female | 23,225 (35·84) | 244 (24·55) | 23,469 (35.67) | 38,014 (41·97) | 172 (33·53) | 38,186 (41.92) |
| Male | 40,388 (62·33) | 742 (74·65) | 41,130 (62.52) | 52,075 (57·49) | 341 (66·47) | 52,416 (57.54) |
| Missing | 1,183 (1·83) | 8 (0·80) | 1,191 (1.81)‡ | 487 (0·54) | 0 (0·00) | 487 (0.53)‡ |
| **Education** | | | | | | |
| None | 3,983 (6·15) | 55 (5·53) | 4,038 (6.14) | 2,842 (3·14) | 13 (2·53) | 2,855 (3.13) |
| Primary | 1,355 (2·09) | 21 (2·11) | 1,376 (2.09) | 1,403 (1·55) | 2 (0·39) | 1,405 (1.54) |
| Secondary | 5,530 (8·53) | 128 (12·88) | 5,658 (8.60) | 4,750 (5·24) | 39 (7·60) | 4,789 (5.26) |
| Tertiary | 15,474 (23·88) | 263 (26·46) | 15,737 (23.92) | 18,239 (20·14) | 173 (33·72) | 18,412 (20.21) |
| Missing | 38,454 (59·35) | 527 (53·02) | 38,981 (59.25) ‡ | 63,342 (69·93) | 286 (55·75) | 63,628 (69.85)‡ |
| **Geopolitical zone** | | | | | | |
| South-west | 31,610 (48·78) | 262 (26·36) | 31,872 (48.45) | 39,110 (43·18) | 124 (24·17) | 39,234 (43.07) |
| South-south | 7,955 (12·28) | 253 (25·45) | 8,208 (12.48) | 8,597 (9·49) | 102 (19·88) | 8,699 (9.55) |
| South-east | 4,263 (6·58) | 93 (9·36) | 4,356 (6.62) | 4,647 (5·13) | 37 (7·21) | 4,684 (5.14) |
| North-central | 12,160 (18·77) | 161 (16·20) | 12,321 (18.73) | 23,728 (26·20) | 140 (27·29) | 23,868 (26.20) |
| North-west | 6,040 (9·32) | 138 (13·88) | 6,178 (9.39) | 10,622 (11·73) | 80 (15·59) | 10,702 (11.75) |
| North-east | 2,768 (4·27) | 87 (8·75) | 2,855 (4.34)‡ | 3,872 (4·27) | 30 (5·85) | 3,902 (4.28) |
| **Symptomatic status** | | | | | | |
| Asymptomatic | 48,928 (75·51) | 350 (35·21) | 49,278 (74.90) | 72,378 (79·91) | 218 (42·50) | 72,596 (79.70) |
| Symptomatic | 15,868 (24·49) | 644 (64·79) | 16,512 (25.10)‡ | 18,198 (20·09) | 295 (57·50) | 18,493 (20.30)‡ |
| **Hospitalisation** | | | | | | |
| No | 30,543 (47·14) | 425 (42·76) | 30,968 (47.07) | 49,213 (54·33) | 171 (33·33) | 49,384 (54.22) |
| Yes | 6,548 (10·11) | 314 (31·59) | 6,862 (10.43) | 1,132 (1·25) | 102 (19·88) | 1,234 (1.35) |
| Missing | 27,705 (42·76) | 255 (25·65) | 27,960 (42.50)‡ | 40,231 (44·42) | 240 (46·78) | 40,471 (44.43)‡ |

CFR: Case Fatality Ratio.

‡ = p-value <0.001.

*: Values may differ from those reported by the NCDC due to the study eligibility criteria.

South-south [41·02; 33·79–49·81 per 100,000 person-days], respectively. The lowest mortality rate was recorded in the South-west for both waves: 28·69 (25·41–32·38) per 100,000 person-days in the first wave and 10·66 (8·94–12·72) per 100,000 person-days in the second wave (see S1 Table for the incidence rate of COVID-19 mortality for individual States of each geopolitical zone). Hospitalised patients recorded a higher mortality rate in both waves, with a higher rate of death recorded in the second wave [307·39; 253·17–373·22 per 100,000 person-days] than in the first wave [198·87; 178·05–222·13 per 100,000 person-days].

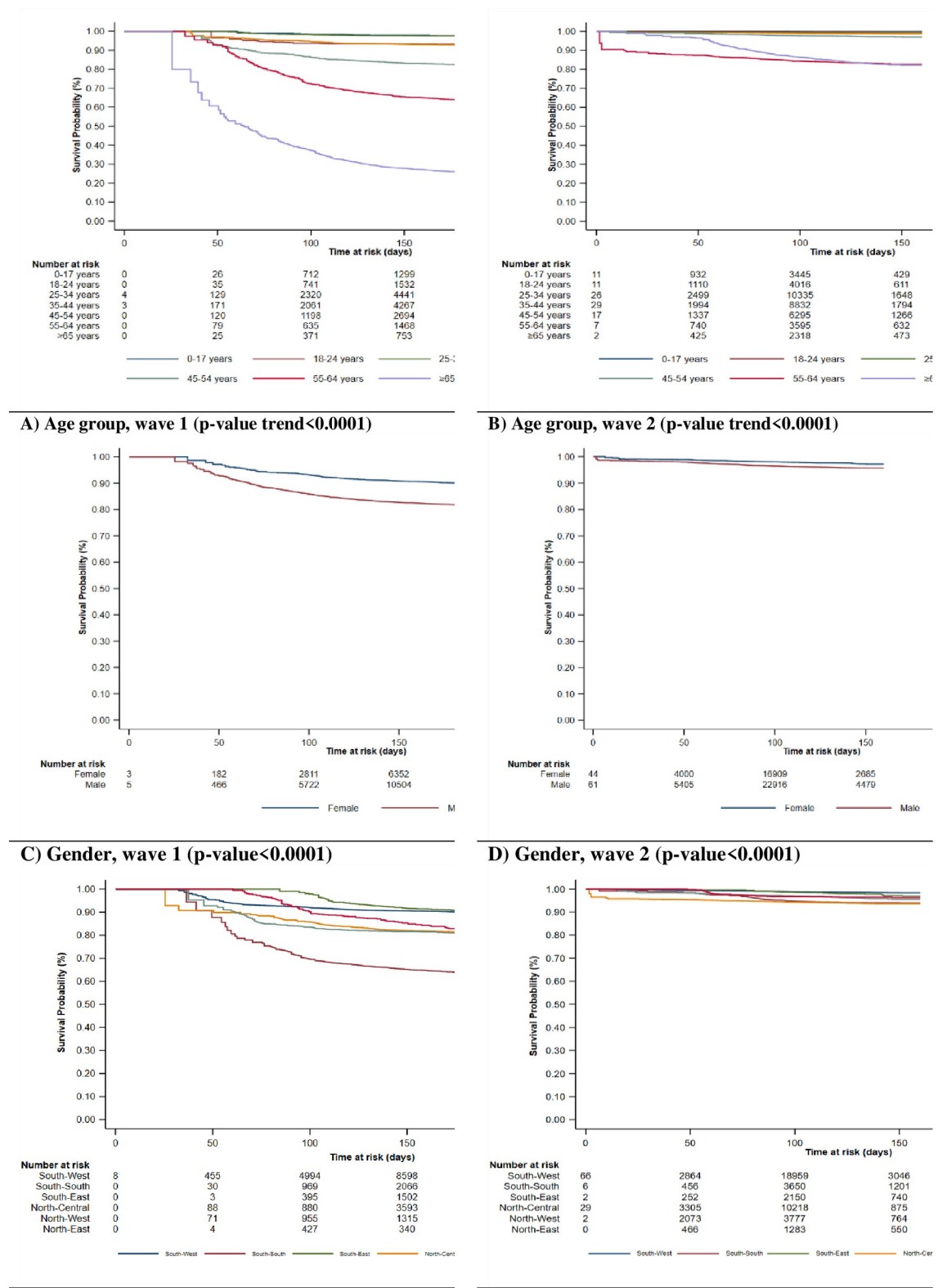

**A) Age group, wave 1 (p-value trend<0.0001)**

**B) Age group, wave 2 (p-value trend<0.0001)**

**C) Gender, wave 1 (p-value<0.0001)**

**D) Gender, wave 2 (p-value<0.0001)**

**E) Geopolitical zone, wave 1 (p-value<0.0001)**

**F) Geopolitical zone, wave 2 (p-value<0.0001)**

**Fig 2. Kaplan Meier plots showing the survival patterns of COVID-19 patients by age group, gender, and geopolitical zone in wave 1 and wave 2.**

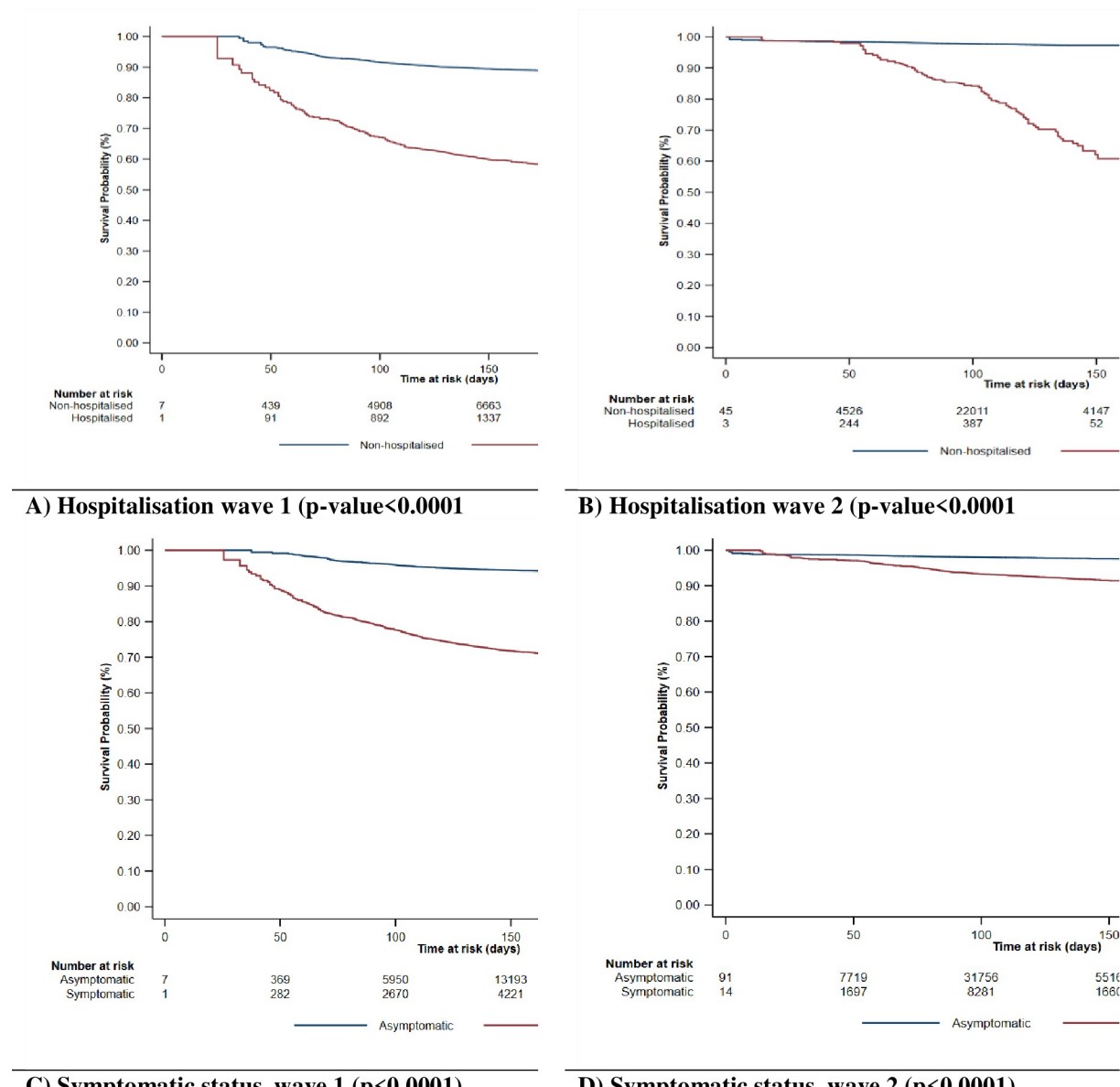

A) Hospitalisation wave 1 (p-value<0.0001

B) Hospitalisation wave 2 (p-value<0.0001

C) Symptomatic status, wave 1 (p<0.0001)

D) Symptomatic status, wave 2 (p<0.0001)

**Fig 3. Kaplan Meier plots showing the survival patterns of COVID-19 patients by hospitalisation and symptomatic status in wave 1 and wave 2.**

## Geospatial distribution of incidence and mortality rates of COVID-19 in the first and second waves

The outputs in Fig 4 illustrates the geospatial burden of COVID-19 reporting the incidence and mortality rates by LGAs (within States) in Nigeria. In terms of incidence rates, the LGAs situated in the South-west States (e.g., LA (Lagos), OG (Ogun), OY (Oyo) and ED (Edo)), as well as LGAs in States in the central part of Nigeria (e.g., KD (Kaduna), FCT (Federal Capital Territory), NA (Nasarawa), PL (Plateau) and BA (Bauchi)) have incidence rates of COVID-19 either range between 101–500 per 100,000, and four LGAs exceeding 500 per 100,000. In terms of incidence rates, we observed that the LGAs with the highest burden in the first wave (Fig 4 (top-left)) were Eti-Osa (cases: 5,854, incidence rate: 1,136.03 per 100,000) followed by

**Table 3. COVID-19 mortality rates during the first and second wave in Nigeria.**

| Variable | Wave 1 (n = 65,524 patients) | | | Wave 2 (n = 90,390 patients) | | |
|---|---|---|---|---|---|---|
| | Death | Person-days at risk | Mortality rate (95% CI) per 100,000 person-days | Death | Person-days at risk | Mortality rate (95% CI) per 100,000 person-days |
| **Overall rate** | 994 | 1,832,290 | **54·25 (50·98–57·73)** | 513 | 2,673,140 | **19·19 (17·60–20·93)** |
| **Age group (year)** | | | | | | |
| 0–17 | 15 | 144,330 | 10·39 (6·27–17·24) | 4 | 237,840 | 1·68 (0·63–4·48) |
| 18–24 | 21 | 163,480 | 12·85 (8·38–19·70) | 6 | 273,640 | 2·19 (0·99–4·88) |
| 25–34 | 51 | 467,110 | 10·92 (8·30–14·37) | 15 | 671,780 | 2·23 (1·35–3·70) |
| 35–44 | 103 | 450,930 | 22·84 (18·83–27·71) | 33 | 589,600 | 5·60 (3·98–7·87) |
| 45–54 | 174 | 284,170 | 61·23 (52·78–71·04) | 70 | 413,820 | 16·92 (13·38–21·38) |
| 55–64 | 257 | 149,990 | 171·34 (151·62–193·63) | 136 | 239,010 | 56·90 (48·10–67·31) |
| ≥65 | 338 | 77,950 | **433·59 (389·75–482·37)** | 222 | 154,490 | **143·70 (125·98–163·90)** |
| Missing | 35 | 94,330 | 37·10 (26·64–51·68) | 27 | 92,950 | 29·05 (19·92–42·36) |
| **Sex** | | | | | | |
| Female | 244 | 656,720 | 37·16 (32·77–42·12) | 172 | 1,121,230 | 15·34 (13·21–17·81) |
| Male | 742 | 1,140,530 | **65·06 (60·54–69·91)** | 341 | 1,537,460 | **22·18 (19·95–24·66)** |
| Missing | 8 | 35,050 | 22·83 (11·42–45·65) | 0 | 14,440 | - |
| **Education** | | | | | | |
| None | 55 | 115,050 | 47·80 (36·70–62·26) | 13 | 81,400 | 15·97 (9·27–27·50) |
| Nursery/primary | 21 | 37,070 | 56·65 (36·94–86·89) | 2 | 40,800 | 4·90 (1·23–19·60) |
| Secondary | 128 | 152,850 | **83·74 (70·42–99·58)** | 39 | 138,760 | 28·11 (20·53–38·47) |
| Tertiary | 263 | 434,890 | 60·48 (53·59–68·24) | 173 | 535,940 | **32·28 (27·81–37·47)** |
| Missing | 527 | 1,092,430 | 48·24 (44·29–52·54) | 286 | 1,876,240 | 15·24 (13·58–17·12) |
| **Geopolitical zone** | | | | | | |
| South-west | 262 | 913,370 | 28·69 (25·41–32·38) | 124 | 1,162,860 | 10·66 (8·94–12·72) |
| South-south | 253 | 214,060 | **118·19 (104·49–133·69)** | 102 | 248,640 | **41·02 (33·79–49·81)** |
| South-east | 93 | 123,420 | 75·35 (61·49–92·33) | 37 | 136,640 | **27·08 (19·62–37·37)** |
| North-central | 161 | 344,220 | 46·77 (40·08–54·59) | 140 | 705,670 | 19·84 (16·81–23·41) |
| North-west | 138 | 168,750 | 81·78 (69·21–96·63) | 80 | 308,250 | 25·95 (20·85–32·31) |
| North-east | 87 | 68,470 | **127·06 (102·98–156·77)** | 30 | 111,090 | 27·00 (18·88–38·62) |
| **Symptomatic status** | | | | | | |
| Asymptomatic | 350 | 1,384,980 | 25·27 (22·76–28·06) | 218 | 2,142,850 | 10·17 (8·91–11·62) |
| Symptomatic | 644 | 447,310 | **143·97 (133·27–155·53)** | 295 | 530,290 | **55·63 (49·63–62·35)** |
| **Hospitalisation** | | | | | | |
| No | 425 | 865,030 | 49·13 (44·68–54·03) | 171 | 1,447,420 | 11·81 (10·17–13·72) |
| Yes | 314 | 157,890 | **198·87 (178·05–222·13)** | 102 | 33,180 | **307·39 (253·17–373·22)** |
| Missing | 255 | 809,370 | 31·51 (27·87–35·62) | 240 | 1,192,530 | 20·13 (17·73–22·84) |

The highest recorded mortality rate is **bold** for each characteristic (the two highest mortality rates are **bolded** for the geopolitical zone).

Lagos Mainland (cases: 4,272, incidence rate: 860.48 per 100,000) which are both located in the state of Lagos. The burden of COVID-19 in wave 2 intensifies (Fig 4 (top-right)), with more LGAs from States in the South-west and Central parts of Nigeria having incidence rates entering ranges of 101–500 per 100,000, and more (i.e., five LGAs) exceeding 500 per 100,000. In terms of mortality rates, while the burden of mortality is marginal across the country; however, it is concentrated in southern and central region of Nigeria. As shown in wave 1, the following states most affected where mortalities are clustered in LGAs were LA (Lagos), ED (Edo), DE (Delta), EB (Ebonyi), FCT (Federal Capital Territory) and NA (Nasarawa) (Fig 4 (bottom-

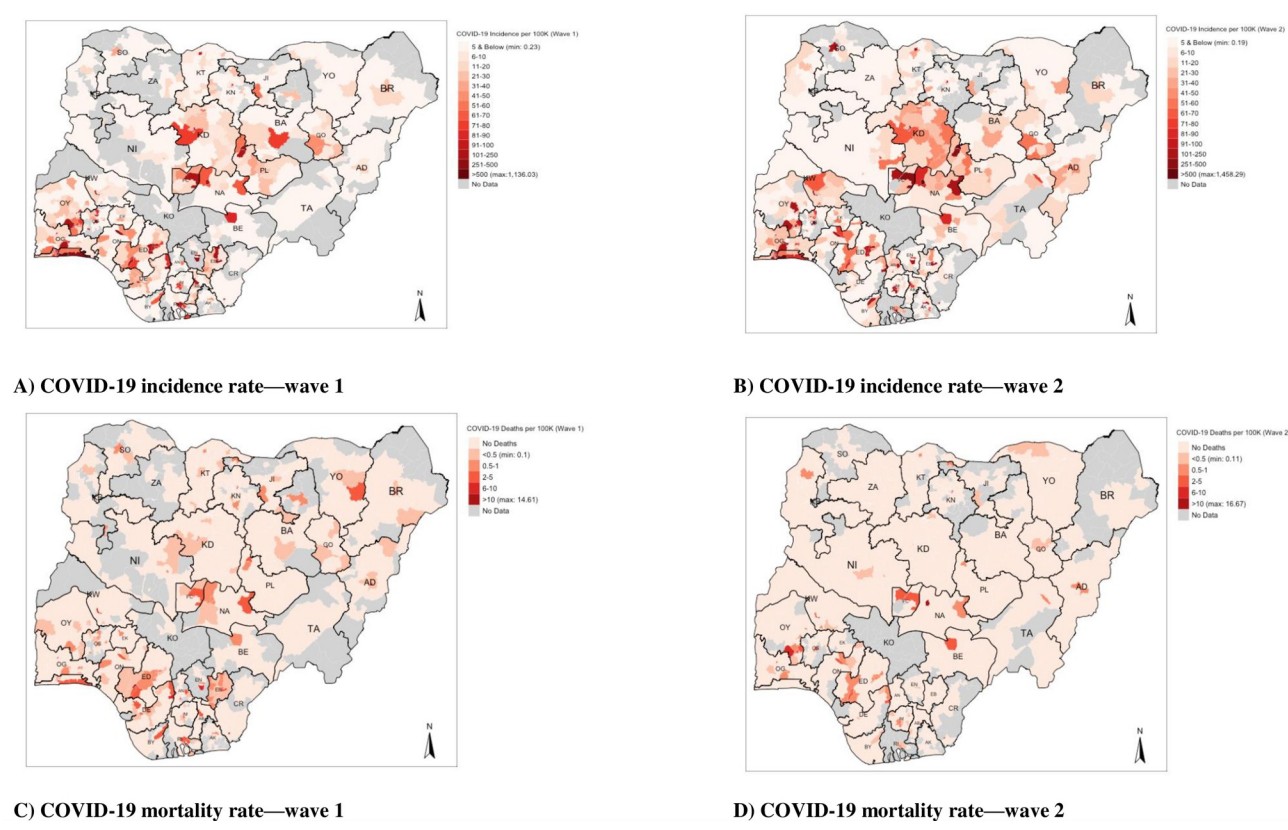

**Fig 4. A Nigerian map showing COVID-19 incidence and mortality rates during the first and second waves.**

left)). Note that the standalone LGA with the greatest burden was Gombe (in the State of Gombe (GO)) with a mortality rate of 14.07 per 100,000. In wave 2, these patterns were diminished, however, the clusters only remained in LGAs for FCT (Federal Capital Territory), ED (Edo) and increased in OY (Oyo) (see Fig 4 (bottom-right)). The standalone LGA with the greatest burden in wave 2 was Keffi (in Nasarawa (NA)) with a mortality rate of 16.57 per 100,000.

Fig 5 compares changes between Wave 2 and Wave 1 to show where there is a growing burden of COVID-19 in Nigeria–where data are available, we can see that LGAs in the following states: NA (Nasarawa), KD (Kaduna), FCT (Federal Capital Territory), AD (Adamawa), KW (Kwara), KB (Kebbi) and OG (Ogun) have an increased incidence rate of COVID-19. However, for mortality rates, there is a broad decrease in such rates, but it should be noted that LGAs in the southern part of OY (State of Oyo) show a marginal increase in the mortality due to COVID-19 where such rates range 1–10 per 100,000.

## Socio-demographic and clinical characteristics of patients associated with COVID-19 mortality in the first and second waves

The unadjusted IRR showed that all patient characteristics (age group, gender, education, geo-political zone, symptomatic status, and hospitalisation) were significantly associated with COVID-19 mortality in both waves (Table 4). The risk of COVID-19 mortality generally increased with increasing age group in both waves, especially among patients aged 35 years or older compared with children (0–17 years). Compared to female patients, the IRR in male

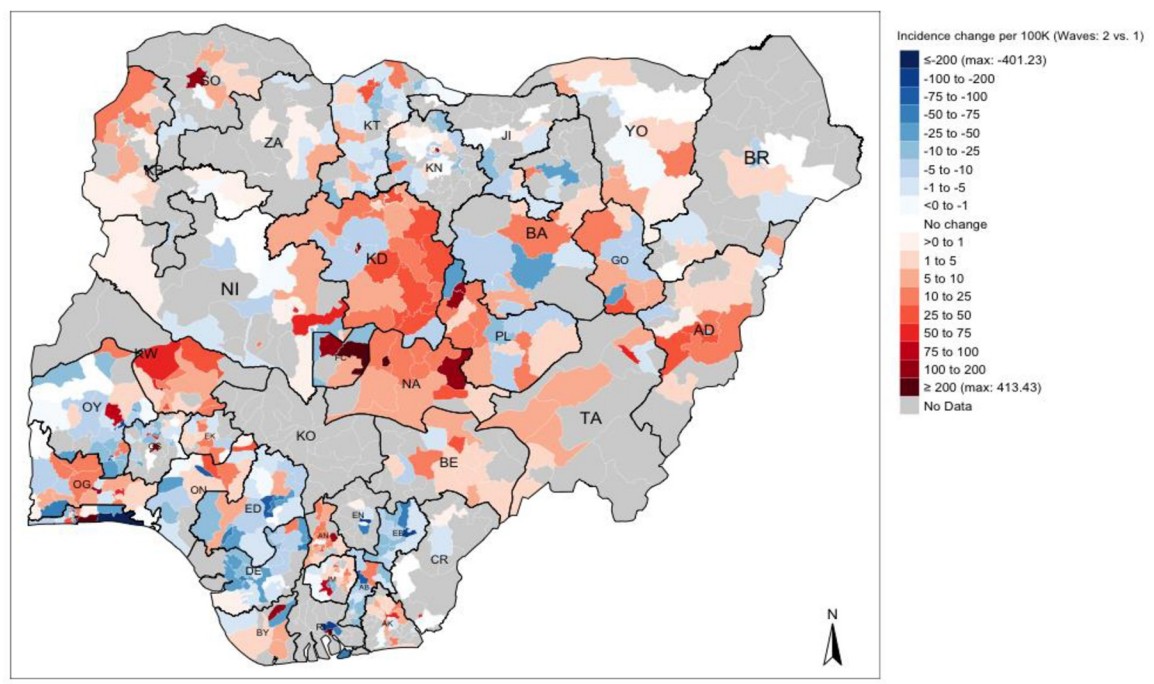

A)

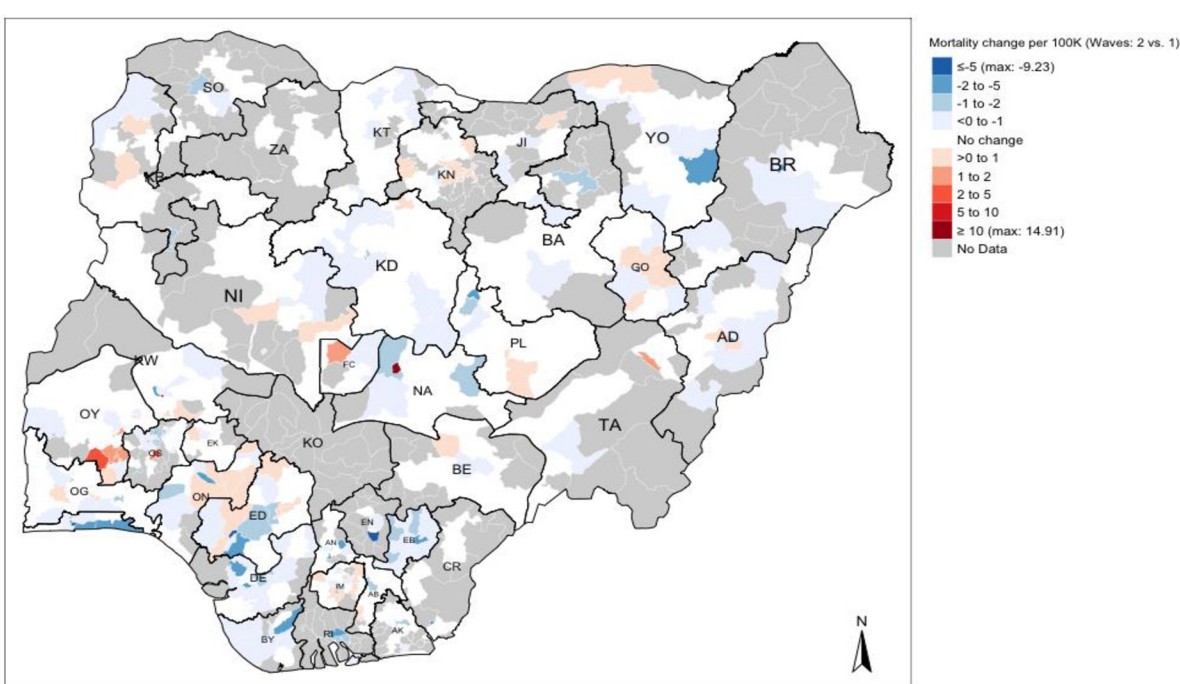

B)

**Fig 5. A Nigerian map showing the changes in COVID-19 incidence and mortality rates during the first and second waves.**

**Table 4. Negative binomial regression modelling of COVID-19 mortality rate ratios during the first and second wave in Nigeria.**

| Variable | Wave 1† | | | | Wave 2 | | | |
|---|---|---|---|---|---|---|---|---|
| | Unadjusted incidence rate ratios (95% CI) | LRT p-value | Adjusted incidence rate ratios (95% CI)† | LRT p-value | Unadjusted incidence rate ratios (95% CI) | LRT p-value | Adjusted incidence rate ratios (95% CI)† | LRT p-value |
| **Age group (year)** | | | | | | | | |
| 0–17 | 1·00 | <0·0001 | 1·00 | <0·0001 | 1·00 | <0·0001 | 1·00 | <0·0001 |
| 18–24 | 1·38 (0·64–3·00) | | 1·30 (0·65–2·61) | | 1·08 (0·25–4·62) | | 1·02 (0·27–3·86) | |
| 25–34 | 1·58 (0·79–3·18) | | 1·22 (0·66–2·26) | | 1·56 (0·43–5·65) | | 1·13 (0·35–3·68) | |
| 35–44 | 3·70 (1·91–7·20) | | **2·68 (1·50–4·79)** | | 4·56 (1·35–15·41) | | 3·05 (0·99–9·35) | |
| 45–54 | 10·36 (5·44–19·73) | | **6·80 (3·87–11·94)** | | 19·22 (5·90–62·66) | | **9·80 (3·30–29·09)** | |
| 55–64 | 27·18 (14·41–51·25) | | **16·55 (9·50–28·84)** | | 60·47 (19·14–191·11) | | **29·19 (10·04–84·88)** | |
| ≥65 | 75·12 (40·01–141·01) | | **37·81 (21·78–65·63)** | | 124·29 (39·74–388·71) | | **52·93 (18·33–152·91)** | |
| **Sex** | | | | | | | | |
| Female | 1·00 | <0·0001 | 1·00 | <0·0001 | 1·00 | 0·007 | 1·00 | 0·0214 |
| Male | 2·41 (1·79–3·24) | | 1·59 (1·33–1·89) | | 1·78 (1·17–2·71) | | 1·36 (1·02–1·80) | |
| **Education** | | | | | | | | |
| None | 1·00 | 0·0014 | 1·00 | 0·0001 | 1·00 | <0·0001 | 1·00 | 0·0005 |
| Nursery/Primary | 1·05 (0·51–2·15) | | 1·42 (0·83–2·44) | | 0·28 (0·05–1·56) | | 0·50 (0·11–2·38) | |
| Secondary | **2·37 (1·38–4·06)** | | **1·68 (1·17–2·40)** | | 2·38 (0·99–5·68) | | **2·58 (1·25–5·34)** | |
| Tertiary | 1·17 (0·70–1·95) | | 0·96 (0·69–1·35) | | 5·35 (2·38–12·06) | | **2·51 (1·29–4·88)** | |
| **Geopolitical zone** | | | | | | | | |
| South-west | 1·00 | 0·0021 | 1·00 | <0·0001 | 1·00 | 0·0109 | 1·00 | 0·0483 |
| South-south | 1·38 (0·89–2·15) | | **1·34 (1·03–1·74)** | | 0·94 (0·50–1·80) | | **1·84 (1·16–2·91)** | |
| South-east | 2·03 (1·25–3·29) | | **1·82 (1·36–2·43)** | | 0·32 (0·15–0·69) | | 1·18 (0·69–2·03) | |
| North-central | 0·86 (0·55–1·34) | | 1·25 (0·96–1·61) | | 0·42 (0·22–0·78) | | **1·67 (1·10–2·54)** | |
| North-west | 1·20 (0·75–1·93) | | **1·80 (1·38–2·33)** | | 0·65 (0·35–1·21) | | **1·81 (1·17–2·81)** | |
| North-east | 1·97 (1·20–3·25) | | **3·11 (2·31–4·19)** | | 0·50 (0·25–1·02) | | 1·55 (0·89–2·71) | |
| **Symptomatic status** | | | | | | | | |
| Asymptomatic | 1·00 | <0·001* | 1·00 | <0·001* | 1·00 | <0·001* | 1·00 | <0·001* |
| Symptomatic | 6·58 (5·06–8·57) | | **3·64 (3·05–4·34)** | | 5·70 (3·91–8·30) | | **2·86 (2·12–3·86)** | |
| **Hospitalisat ion** | | | | | | | | |
| No | 1·00 | <0·0001 | 1·00 | <0·0001 | 1·00 | <0·0001 | 1·00 | <0·0001 |
| Yes | 5·61 (4·09–7·71) | | **2·97 (2·40–3·67)** | | 18·67 (11·28–30·89) | | **9·45 (6·24–14·29)** | |

†: All Incidence Rate Ratios are mutually adjusted for all other variables in the table.

LRT: Likelihood ratio test

*: Wald's p-value.

Statistically significant results (p≤0·05) are in **bold.**

IRR values of variables (e.g., sex and hospitalisation) with missing categories were excluded in this table.

patients was two-fold higher [IRR 2·41; 95% CI: 1·79–3·24] in the first wave and 78% higher [IRR 1·78; 95% CI: 1·17–2·71] in the second wave. In both waves, patients with secondary and tertiary education had an increased risk of COVID-19 mortality compared to the uneducated at diagnosis. Except for the North-central, the IRR for COVID-19 mortality in the first wave was higher in patients in other geopolitical zones than those in the South-west. Although not statistically significant, a reverse trend was noted in the second wave in that patients in all the

zones recorded a lower risk of mortality than those in the South-west. The IRR in symptomatic patients in the first wave was about 7 times higher [IRR 6·58; 5·06–8·57] than in asymptomatic patients at testing; a similar trend was noted in the second wave. Likewise, the risk of COVID-19 mortality in hospitalised patients was significantly higher than in non-hospitalised patients in both waves.

Apart from geopolitical zone that was marginally statistically significant (p = 0·0483) in the second wave, all the patients' characteristics remained significantly associated with COVID-19 mortality in both waves in the adjusted model. Except for patients aged 18 to 34 years, the risk of COVID-19 mortality in the first wave was higher in patients aged 35 years or older than in children. A similar trend was observed in the second wave, except the age group at risk of COVID-19 mortality increased to 45 years or older. Compared to female patients, the adjusted IRR for male patients was 59% [95% CI: 1·33–1·89] higher in the first wave and 36% [95% CI: 1·02–1·80] higher in the second wave. The risk of COVID-19 mortality in symptomatic patients was 4 times higher [aIRR 3·64; 3·05–4·34] in the first wave and 3 times higher [aIRR 2·86; 2·12–3·86] in the second wave, compared to asymptomatic patients at testing. Compared to non-hospitalised patients, hospitalisation remained significantly associated with a higher risk of COVID-19 mortality, both in the first [aIRR 2·97; 2·40–3·67] and second [aIRR 9·45; 6·24–14·29] waves. Patients with secondary education remained at higher risk of COVID-19 mortality than the uneducated in the first wave, as were those with tertiary education in the second wave.

## Discussion

### Summary and interpretation of key findings

The COVID-19 mortality rate was substantially higher in the first wave [54·25 per 100,000 person-days; 95% CI: 50·98–57·73] than in the second wave [19·19; 95% CI: 17·60–20·93]. Patients who were older, male, symptomatic at diagnosis and required hospitalisation were at higher risk of COVID-19 mortality during both waves. To date, there is limited evidence on the use of the measures of effect in our study to facilitate a direct comparison of findings. Therefore, it is difficult to categorically state that the incidence rates of COVID-19 mortality in the present study are low or high. However, a Mexican study reported a mortality rate of 4·94 per 1000 person-years [22], which is higher than our values. This could be due to the sole focus of the Mexican study on symptomatic COVID-19 patients only, as opposed to ours that focused on both symptomatic and asymptomatic patients. In South Africa, the incidence rates of in-hospital deaths in both first (3·6 deaths per 100,000 people) and second (8·3 deaths per 100,000) waves appear to be lower than those in our study [23], despite our focus being on both hospitalised and non-hospitalised patients. Overall, the 30-day COVID-19 mortality rate suggest that there were not a lot of COVID-19 patients at risk of death, as indicated by the survival curves which did not reach the value of zero, albeit the patients who died did so at a very high rate. Of note, the COVID-19 mortality trends identified in Nigeria could be related to the under-detection of cases. This is because, as of writing, Nigeria has conducted 14.87 tests per 1,000 population, with Ethiopia conducting 30.39 tests per 1,000 population and South Africa conducting 300.71 tests per 1,000 population [5]. Outside Africa, the US, UK, and Argentina, respectively, have conducted 1,792.22, 4,193.57 and 531.85 tests per population during the same period [5].

The higher incidence rate of COVID-19 mortality in the first wave compared to the second wave in the present study could be attributable to several factors. First, home-based care in Nigeria was rolled out towards the end of the first wave, which became well established in the second wave. Home-based care in the Nigerian context is when a person confirmed to have

COVID-19 is provided with required medical care at home by a family member, a friend, or an identified person, with clinical advice and support from designated health workers [16]. It is, therefore, possible that a higher triage threshold was applied for hospitalisation, resulting in fewer but more severe patients being hospitalised.

Concerning hospitalisation among COVID-19 deaths and survivors, 10% of survivors were hospitalised in wave 1, while only 1% in wave 2. This substantial difference could be attributable to improvement in case management, including ambulatory care, such that very few COVID-19 patients needed hospitalisation. However, despite the minimal difference in the proportion of missing data for this patient group in both waves (43% in the first wave and 44% in the second wave), under-reporting of COVID-19 cases in the second wave is a reasonable explanation and worth considering. Another notable difference between the two waves is how much worse hospitalised patients seemed to do during the second wave compared to the first wave. Increased in-hospital mortality in the second wave in South Africa was attributed to the admission of older patients, increased health system pressure and predominance of SARS-CoV-2 Beta lineage [23]. This is similar to the characteristics of patients hospitalised in the second wave in Nigeria, with a significant proportion presenting with severe illness, comorbidities and needing specialised medical care [Personal Communication with COVID-19 Treatment Centres' Manager]. Although there were fewer hospitalisations and more deaths in the second wave, the observed trend has important implications for public health in Nigeria. It potentially implies a need to prioritise the integration of health security and universal health coverage, which currently run separately in Nigeria. The finding also underlines a gap in ability of existing healthcare systems to deliver life-saving treatment when resources are stretched by a surging pandemic or circulation of a more dangerous variant of SARS-CoV-2. The use of genomic data to identify which SARS-CoV-2 strains predominated in Nigeria's first and second waves would have aided the understanding of these findings. However, African countries, including Nigeria, have contributed few SARS-CoV-2 genomic data towards the global pool [24].

Unsurprisingly, COVID-19 patients asymptomatic at testing accounted for about half of deaths in the second wave but had a much lower mortality rate (10·17 per 100,000 person-days) than symptomatic patients (55·63 per 100,000 person-days) during this wave. The positive association between being symptomatic at diagnosis and increased rate and risk of COVID-19 mortality in both waves in the present study is also not surprising. Compared with asymptomatic patients at testing, those who present with COVID-19 symptoms tend to be older and present with a high prevalence of comorbidities, characteristics shown to be associated with increased risk of COVID-19-related mortality [25–27]. Furthermore, the finding of male patients being at higher risk of COVID-19 mortality than their female counterparts in the present study is congruent with existing literature [9, 28]. Possible explanations for the higher susceptibility of males over females include hormonal differences and gender-specific lifestyle [29], as well as the burden of comorbidities [30]. We found geographic differences in the rate and risk of COVID-19 mortality in both waves. Notably, patients in the South-west region of Nigeria had the lowest mortality rates in both waves. This finding could be attributed mainly to the location and importance of Lagos State (the epicentre of COVID-19 in the country) in the region. Lagos is the most populated urban state and a central domestic and international travel hub in Nigeria [31]. For example, Lagos Murtala Muhammed International Airport remained the busiest airport for both international and domestic travellers in 2018, serving at least 3.5 million passengers (48.7% of total air passengers across Nigeria) in the first half of 2018; 41.8% of these passengers were international passengers [32]. As such, Lagos State promptly conducted a risk assessment for COVID-19 and based on the assessment outcomes, instituted appropriate public health interventions, including training and re-training public

health personnel in case management, risk communications and equipping laboratories and treatment centres with relevant supplies.

Conversely, there was a high incidence and risk of COVID-19 mortality in the North-east, despite accounting for the lowest absolute mortality counts in both waves. This could be that there was low testing capacity in the state and the likelihood is that the people who were tested and therefore, included in the SORMAS database had more severe illness leading to more deaths in the state. Adamawa is an example of such a state that relies on sending samples to neighbouring States or the NCDC National Reference Laboratory in Abuja. It is also worth noting that Kogi State in North-central Nigeria contributed minimal data to the analyses (S1 Table) due to governmental policies.

The higher risk of COVID-19 mortality among educated patients compared to the less educated patients could be explained by several factors. First, it is possible that compared with less-educated COVID-19 patients, those with a higher level of education are more likely to lead a sedentary lifestyle due to the nature of their jobs, be overweight/obese, and have comorbidities [33]; these have been identified as risk factors for COVID-19 mortality [34]. Second, patients with a higher level of education tend to be more internationally mobile which puts them at greater risk of COVID-19 infection than their less-educated counterparts. Moreover, educated patients, who are more likely to reside in more structured settings for ease of contact tracing, may be more likely to be followed up and therefore have their clinical outcome ascertained than their less-educated counterparts (who are more likely to reside in peri-urban settings).

## Study strengths, limitations, and generalisability

To the best of our knowledge, this large, retrospective cohort study is the first in Nigeria (and possibly in an African setting) to have estimated the incidence rates of COVID-19 mortality and associated risk factors across two epidemiological waves. Findings from this study will therefore be crucial to public health agencies and health facilities in evaluating existing COVID-19 case management strategies and advocating for more investments in surveillance systems. We utilised laboratory-confirmed COVID-19 records, thus minimising misclassification of the denominator population in estimating the incidence rate; however, we lacked data on the validation of COVID-19 deaths by clinicians on SORMAS irrespective of SARS-CoV-2 test outcome. In the absence of clinical confirmation of the cause of mortality, some deaths in the present study might not be COVID-19 related, thus leading to an overestimation of COVID-19 mortality. Our dataset was obtained via the IDSR platform, meaning data from all the levels of governance (federal, state, and local), including those from persons managed at home had the opportunity of being captured by the surveillance system. However, our evidence using data from SORMAS directly depends on the strength of routine national surveillance data collection. Therefore, the mixture of active and passive nature of COVID 19 testing in Nigeria implies that our findings may not fully represent the COVID-19 situation in the country, especially at the community level and during the second wave when a substantial decline in contact tracing was noted. The use of data from SORMAS also limited our capacity to explore additional risk factors for COVID-19 mortality including comorbidities (hypertension and diabetes) [35].

Additionally, we lacked sufficient variables to ascertain the significant association between socioeconomic status (measured by the Distressed Communities Index (DCI) and its components) on COVID-19-related mortality [36]. However, although the significant effect of educational level has been noted elsewhere [36], weighted categories of education and occupation, as recommended by Ibadin and Akpede [37], might be more feasible for determining

socioeconomic status in Nigeria than DCI. We assumed that mortalities among persons who tested positive for SARS-CoV-2 were COVID-19-related. In addition, our definition of survivors included a loss to follow-ups, some of whom could have died.

In conclusion, the rate of COVID-19 mortality in Nigeria was higher in the first wave than in the second wave. While this could suggest improvement in public health response and care during the second wave, potential limitations of the surveillance data used for this study need to be considered. Further, the regional differences in COVID-19 suggest that policymakers need to address regional equity in access to testing and quality of care to mitigate the deleterious impacts of the ongoing third wave in Nigeria. The findings also underline the importance of prioritising the integration of health security and universal health coverage in a resource-limited setting, such as Nigeria. Lastly, the findings have provided novel and context-specific evidence for interpreting the COVID-19 burden in Nigeria and possibly in other WHO African countries with similar population profiles and surveillance systems.

## Supporting information

**S1 Checklist. STROBE statement.**
(DOC)

**S1 Table. Incidence rates for COVID-19 mortality in the Nigerian States.**
(DOCX)

## Acknowledgments

We are grateful to all the treatment centres and laboratories within the NCDC networks for providing both epidemiological and laboratory data to SORMAS regularly. We express our profound gratitude to the federal and state SORMAS team members for collating and maintaining a quality dataset utilised for this study.

## Author Contributions

**Conceptualization:** Kelly Elimian, Anwar Musah, Ehimario Igumbor, Giulia Gaudenzi, Ekaete Tobin, Stephen Obaro, Tobias Alfven, Chinwe Ochu.

**Data curation:** Olaolu Aderinola, Cyril Erameh, William Nwanchukwu, Oladipo Ogunbode, Ismail Abdus-Salam, Yahya Disu, Cyprian Oshoma, Chinedu Arinze, Blessing Ebhodaghe, Chioma Dan-Nwafor, Emmanuel Pembi, Ifeanyi Ike, Bamidele Mutiu, Obinna Nwafor, Mildred Okowa, Sebastian Yennan, David Olatunji, Lanre Falodun, Emmanuel Joseph, Ifeanyi Abali, Tarik Mohammed, Benjamin Yiga, Khadeejah Kamaldeen, Nwando Mba, John Oladejo, Elsie Ilori, Olusola Aruna, Khadeejah Hamza, Michael Asuzu, Friday Okonofua, Yusuf Deeni, Chinwe Ochu, Chikwe Ihekweazu.

**Formal analysis:** Kelly Elimian, Anwar Musah, Carina King, Puja Myles, Olubunmi Olopha, Cyprian Oshoma.

**Investigation:** Kelly Elimian, Ehimario Igumbor, Olaolu Aderinola, Cyril Erameh, William Nwanchukwu, Oluwatosin Akande, Oladipo Ogunbode, Abiodun Egwuenu, Ismail Abdus-Salam, Olubunmi Olopha, Yahya Disu, Abimbola Bowale, Cyprian Oshoma, Cornelius Ohonsi, Chinedu Arinze, Sikiru Badaru, Blessing Ebhodaghe, Zaiyad Habib, Michael Olugbile, Chioma Dan-Nwafor, Jafiya Abubakar, Emmanuel Pembi, Lauryn Dunkwu, Ifeanyi Ike, Ekaete Tobin, Bamidele Mutiu, Rejoice Luka-Lawal, Obinna Nwafor, Mildred Okowa, Chidiebere Ezeokafor, Emem Iwara, Sebastian Yennan, Sunday Eziechina, Lanre Falodun, Emmanuel Joseph, Ifeanyi Abali, Tarik Mohammed, Benjamin Yiga, Khadeejah

Kamaldeen, Emmanuel Agogo, Nwando Mba, John Oladejo, Elsie Ilori, Olusola Aruna, Geoffrey Namara, Khadeejah Hamza, Shaibu Bello, Friday Okonofua, Yusuf Deeni, Ibrahim Abubakar, Chinwe Ochu, Chikwe Ihekweazu.

**Methodology:** Kelly Elimian, Anwar Musah, Carina King, Ehimario Igumbor, Puja Myles, Cyril Erameh, William Nwanchukwu, Oluwatosin Akande, Ndembi Nicaise, Oladipo Ogunbode, Abiodun Egwuenu, Emily Crawford, Giulia Gaudenzi, Olubunmi Olopha, Yahya Disu, Abimbola Bowale, Sikiru Badaru, Blessing Ebhodaghe, Zaiyad Habib, Emmanuel Pembi, Lauryn Dunkwu, Ifeanyi Ike, Ekaete Tobin, Mildred Okowa, Chidiebere Ezeokafor, David Olatunji, Tarik Mohammed, Emmanuel Agogo, Yusuf Deeni, Ibrahim Abubakar, Tobias Alfven, Chinwe Ochu.

**Project administration:** Kelly Elimian, Olaolu Aderinola, Cyril Erameh, William Nwanchukwu, Oluwatosin Akande, Ndembi Nicaise, Oladipo Ogunbode, Emily Crawford, Ismail Abdus-Salam, Olubunmi Olopha, Abimbola Bowale, Cyprian Oshoma, Cornelius Ohonsi, Chinedu Arinze, Sikiru Badaru, Blessing Ebhodaghe, Zaiyad Habib, Michael Olugbile, Chioma Dan-Nwafor, Jafiya Abubakar, Emmanuel Pembi, Lauryn Dunkwu, Ifeanyi Ike, Ekaete Tobin, Bamidele Mutiu, Rejoice Luka-Lawal, Obinna Nwafor, Mildred Okowa, Chidiebere Ezeokafor, Emem Iwara, Sebastian Yennan, Sunday Eziechina, David Olatunji, Lanre Falodun, Emmanuel Joseph, Ifeanyi Abali, Tarik Mohammed, Benjamin Yiga, Khadeejah Kamaldeen, Emmanuel Agogo, Nwando Mba, John Oladejo, Geoffrey Namara, Stephen Obaro, Michael Asuzu, Shaibu Bello, Friday Okonofua, Chikwe Ihekweazu.

**Resources:** Ehimario Igumbor, Ismail Abdus-Salam, Yahya Disu, Abimbola Bowale, Sikiru Badaru, Zaiyad Habib, Chioma Dan-Nwafor, Jafiya Abubakar, Ifeanyi Ike, Bamidele Mutiu, Rejoice Luka-Lawal, Obinna Nwafor, Mildred Okowa, Emem Iwara, Sebastian Yennan, Sunday Eziechina, Lanre Falodun, Emmanuel Joseph, Ifeanyi Abali, Benjamin Yiga, Khadeejah Kamaldeen, Nwando Mba, John Oladejo, Elsie Ilori, Olusola Aruna, Geoffrey Namara, Khadeejah Hamza, Shaibu Bello, Friday Okonofua, Yusuf Deeni, Ibrahim Abubakar, Tobias Alfven, Chikwe Ihekweazu.

**Software:** Kelly Elimian, Anwar Musah, Abiodun Egwuenu, Giulia Gaudenzi, Chinedu Arinze, Lauryn Dunkwu, Ifeanyi Ike, Obinna Nwafor.

**Supervision:** Puja Myles, Olaolu Aderinola, Cyril Erameh, Oladipo Ogunbode, Abiodun Egwuenu, Ismail Abdus-Salam, Olubunmi Olopha, Yahya Disu, Abimbola Bowale, Cornelius Ohonsi, Chinedu Arinze, Sikiru Badaru, Zaiyad Habib, Michael Olugbile, Chioma Dan-Nwafor, Jafiya Abubakar, Emmanuel Pembi, Lauryn Dunkwu, Ekaete Tobin, Bamidele Mutiu, Rejoice Luka-Lawal, Chidiebere Ezeokafor, Emem Iwara, Sebastian Yennan, Sunday Eziechina, Lanre Falodun, Emmanuel Joseph, Ifeanyi Abali, Benjamin Yiga, Khadeejah Kamaldeen, Emmanuel Agogo, John Oladejo, Elsie Ilori, Geoffrey Namara, Michael Asuzu, Shaibu Bello, Yusuf Deeni, Ibrahim Abubakar, Chinwe Ochu, Chikwe Ihekweazu.

**Validation:** Anwar Musah, Olaolu Aderinola, Oluwatosin Akande, Ndembi Nicaise, Abiodun Egwuenu, Emily Crawford, Giulia Gaudenzi, Ismail Abdus-Salam, Yahya Disu, Abimbola Bowale, Cyprian Oshoma, Cornelius Ohonsi, Chinedu Arinze, Sikiru Badaru, Blessing Ebhodaghe, Zaiyad Habib, Michael Olugbile, Chioma Dan-Nwafor, Jafiya Abubakar, Lauryn Dunkwu, Ifeanyi Ike, Bamidele Mutiu, Rejoice Luka-Lawal, Obinna Nwafor, Mildred Okowa, Emem Iwara, Sebastian Yennan, Sunday Eziechina, David Olatunji, Lanre Falodun, Emmanuel Joseph, Ifeanyi Abali, Tarik Mohammed, Benjamin Yiga, Khadeejah Kamaldeen, Emmanuel Agogo, Nwando Mba, John Oladejo, Elsie Ilori, Olusola Aruna,

Geoffrey Namara, Stephen Obaro, Khadeejah Hamza, Michael Asuzu, Shaibu Bello, Friday Okonofua, Yusuf Deeni, Ibrahim Abubakar, Tobias Alfven, Chikwe Ihekweazu.

**Visualization:** Carina King, Ehimario Igumbor, Giulia Gaudenzi, Cyprian Oshoma, Cornelius Ohonsi, Chinedu Arinze, Sikiru Badaru, Lauryn Dunkwu, Rejoice Luka-Lawal, Elsie Ilori, Geoffrey Namara.

**Writing – original draft:** Kelly Elimian, Anwar Musah, Carina King, Puja Myles, Ndembi Nicaise, Emily Crawford.

**Writing – review & editing:** Kelly Elimian, Anwar Musah, Carina King, Ehimario Igumbor, Puja Myles, Olaolu Aderinola, Cyril Erameh, William Nwanchukwu, Oluwatosin Akande, Ndembi Nicaise, Oladipo Ogunbode, Abiodun Egwuenu, Emily Crawford, Giulia Gaudenzi, Ismail Abdus-Salam, Olubunmi Olopha, Yahya Disu, Abimbola Bowale, Cyprian Oshoma, Cornelius Ohonsi, Chinedu Arinze, Sikiru Badaru, Blessing Ebhodaghe, Zaiyad Habib, Michael Olugbile, Chioma Dan-Nwafor, Jafiya Abubakar, Emmanuel Pembi, Lauryn Dunkwu, Ifeanyi Ike, Ekaete Tobin, Bamidele Mutiu, Rejoice Luka-Lawal, Obinna Nwafor, Chidiebere Ezeokafor, Emem Iwara, Sebastian Yennan, Sunday Eziechina, David Olatunji, Lanre Falodun, Emmanuel Joseph, Ifeanyi Abali, Tarik Mohammed, Benjamin Yiga, Khadeejah Kamaldeen, Emmanuel Agogo, Nwando Mba, John Oladejo, Elsie Ilori, Olusola Aruna, Geoffrey Namara, Stephen Obaro, Khadeejah Hamza, Michael Asuzu, Shaibu Bello, Friday Okonofua, Yusuf Deeni, Ibrahim Abubakar, Tobias Alfven, Chinwe Ochu, Chikwe Ihekweazu.

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
