## [Decision Letter · Decision Letter 0]

29 Nov 2021

PGPH-D-21-00823

Incidence of COVID-19 mortality and its associated factors during the first and second waves in Nigeria: a retrospective cohort study

Dear Dr. Elimian,

Thank you for submitting your manuscript to PLOS Global Public Health. After careful consideration, we feel that it has merit but does not fully meet PLOS Global Public Health’s publication criteria as it currently stands. Therefore, we invite you to submit a revised version of the manuscript that addresses the points raised during the review process.

Please consider and address to comments and suggestions made by our external reviewers. As pointed out, the classification of the type of study should be revised, as this study is not a cohort study, which presupposes the tracking and comparison of two or more groups (or coohorts) forward from exposure to outcome.

We look forward to receiving your revised manuscript.

Kind regards,

Everton Falcão de Oliveira, Ph.D

Academic Editor

Journal Requirements:

1. In your ethics statement in the Methods section and in the online submission form, please provide additional information about the data used in your retrospective study. Specifically, please ensure that you have discussed whether all data were fully anonymized before you accessed them and/or whether the IRB or ethics committee waived the requirement for informed consent. If patients provided informed written consent to have data from their medical records used in research, please include this information.

2. Since your data is not available for proprietary reasons, please explain via email why the data is not available. Please also include the contact information for the third party organization that should be contacted should other researchers want to request access to this data and please include the full citation of where the data can be found. We also request that you verify with us via email that any researcher will be able to obtain the data set in the same manner that the you have obtained it. If you feel you are unwilling or unable to adhere to this policy, please explain your reasons by return email and your exemption request will be escalated to the editor for approval. Your exemption request will be handled independently and will not hold up the peer review process, but will need to be resolved should your manuscript be accepted for publication. One of the Editorial team will be in touch if they require more information.

Reviewers' comments:

Reviewer's Responses to Questions

**Comments to the Author**

1. Does this manuscript meet PLOS Global Public Health’s publication criteria? Is the manuscript technically sound, and do the data support the conclusions? The manuscript must describe methodologically and ethically rigorous research with conclusions that are appropriately drawn based on the data presented.

Reviewer #1: Yes

Reviewer #2: Yes

2. Has the statistical analysis been performed appropriately and rigorously?

Reviewer #1: Yes

Reviewer #2: Yes

3. Have the authors made all data underlying the findings in their manuscript fully available (please refer to the Data Availability Statement at the start of the manuscript PDF file)?

Reviewer #1: Yes

Reviewer #2: No

4. Is the manuscript presented in an intelligible fashion and written in standard English?

Reviewer #1: Yes

Reviewer #2: Yes

5. Review Comments to the Author

Reviewer #1: This is a robust study based on surveillance data. Data collection and statistical analyses are described in detail. The use of negative binomial regression is appropriately justified. There are however some aspects that deserve clarification:

1. The authors state that RT PCR was collected upon meeting COVID-19 definitions, yet they classify a great number of subjects as asymptomatics. Besides needing clarification, this aspect poses a question: why include asymptomatic subjects in the cohort? Did they not inflate the relative risk of other co-variables? Maybe factors associated with symptomatic disease are confounding those associated with hospital admission or death.

2. Socioeconomic variables are missing, except for schooling. That is a major limitation, that should be emphasized in the discussion (of course, schooling and geographical data can be proxies of socioeconomic variables, but they are of too much importante, as demonstrated im previous study, so that their absense is a major limitation for inferences and generalization of findings).

3. Since much importance was attibuted to geopolitical zones, the study could be improved by including maps (e.g., Kernel density maps of incidence and mortality)

Reviewer #2: The manuscript "Incidence of COVID-19 mortality and its associated factors during the first and second waves in Nigeria: a retrospective cohort study" mainly estimates mortality and identifies associated mortality risk factors during the first and second waves in Nigeria. Although, I appreciate authors efforts in collecting dataset there are several issues that needs to be addressed before publishing these results.

1- I would first like to invite the authors to reflect on whether the study in question is of the retrospective cohort type. Although at some point in the study, specifically in the survival analysis, it could be a retrospective cohort study (the authors jump back in time to identify a cohort of individuals at a point in time before they had developed the outcome of interest (death by COVID19), and they try to establish their exposure status at that point in time, and they then determine whether the subjects subsequently developed the outcome of interest), I think the objective of the study could be embedded in another type of observational study. 

2- It would be interesting to see on a map the geographic distribution of the CFR and other variables in the 36 States and the Federal Capital Territory (FCT) in Nigeria. Furthermore, it would be very interesting to discuss the geographic pattern found regarding the results on sociodemographic and clinical characteristics (age group, gender, education, geopolitical zone, symptomatic status, and hospitalization), according to the geographic pattern found.

3- I strongly suggest unifying the sections: "data source", "data collection", "data management and viable definition". They also suggested including within this new item Table 1 of the supplementary material (Definition and classification of study covariates). The inclusion of this table is fundamental so that the reader can follow the process described in the analysis.

4- Although the introduction is clear and explains very well the importance of estimating mortality in Nigeria, it is important to introduce the reader to the possible risk factors associated with mortality from COVID-19. This in addition to giving an idea of what will be shown in the article, will support the discussion. 

5- Neither in the title nor in the abstract is it implied that a survival analysis was performed. 

6- In Table 1 indicate in some way the significant differences between the groups. In addition, I suggest clarifying whether the comparison of the proportions (age, sex, education, etc.) was made within groups corresponding to each wave (ie Survivor Wave 1 vs. Death Wave 1) or between groups (ie Survivor Wave 1 vs. Survivor Wave 2; Death Wave 1 vs. Death Wave 2).

7- Similar to the previous point, indicate if the death rate was compared between the first wave and the second wave and also indicate when this difference was significant. 

8- Why did the survival curves not reach the value 0? Arguing in the discussion. 

9- I suggest describing in the introduction and discussing the different prevention measures that were established by the Nigerian government (national or regional) and that could influence the temporal pattern of COVID19 mortality in Nigeria.

6. PLOS authors have the option to publish the peer review history of their article (what does this mean?). If published, this will include your full peer review and any attached files.

**Do you want your identity to be public for this peer review?** For information about this choice, including consent withdrawal, please see our Privacy Policy.

Reviewer #1: **Yes: **CARLOS MAGNO CASTELO BRANCO FORTALEZA

Reviewer #2: No

---

## [Editor Report · Decision Letter 1]

13 Apr 2022

PGPH-D-21-00823R1

COVID-19 mortality rate and its associated factors during the first and second waves in Nigeria

Dear Dr. Elimian,

Thank you for submitting your manuscript to PLOS Global Public Health. After careful consideration, we feel that it has merit but does not fully meet PLOS Global Public Health’s publication criteria as it currently stands. Therefore, we invite you to submit a revised version of the manuscript that addresses the points raised during the review process.

I appreciate the efforts of the authors to address all the reviewer's comments. However, it is necessary that all corrections made in the manuscript are adapted for the abstract. For example, the study type remains incorrect in the abstract. In addition, the abstract does not include all the analyzes performed and the main study results.

We look forward to receiving your revised manuscript.

Kind regards,

Everton Falcão de Oliveira, Ph.D

Academic Editor

Journal Requirements:

1. Your co-authors:

Ismail Abdus-Salam -dradeshina@yahoo.com

Sikiru Badaru -sikiru.badaru@ncdc.gov.ng

Bamidele Mutiu -drmutiu@yahoo.com

Obinna Nwafor -obinna.nwafor@ncdc.gov.ng

Benjamin Yiga -gandibenjamin@yahoo.com

Khadeejah Hamza -deejahhamza@gmail.com

Michael Asuzu -mcasuzu2003@yahoo.com

Yusuf Deeni -yusufdeeni.yd@gmail.com

Ibrahim Abubakar -i.abubakar@ucl.ac.uk

,have not confirmed authorship of the manuscript. We have resent them the authorship confirmation email; however please check that the above email address for them is correct and follow up personally to ensure they confirm. 

Please note that we cannot proceed your manuscript  until we have received confirmations from all co-authors.
---

## [Editor Report · Decision Letter 2]

3 May 2022

COVID-19 mortality rate and its associated factors during the first and second waves in Nigeria

PGPH-D-21-00823R2

Dear Dr Elimian,

We are pleased to inform you that your manuscript 'COVID-19 mortality rate and its associated factors during the first and second waves in Nigeria' has been provisionally accepted for publication in PLOS Global Public Health.

Best regards,

Everton Falcão de Oliveira, Ph.D

Academic Editor